# Flavonoids from *Piper* Species as Promising Antiprotozoal Agents against *Giardia intestinalis*: Structure-Activity Relationship and Drug-Likeness Studies

**DOI:** 10.3390/ph15111386

**Published:** 2022-11-10

**Authors:** Juan C. Ticona, Pablo Bilbao-Ramos, Ángel Amesty, Ninoska Flores, M. Auxiliadora Dea-Ayuela, Isabel L. Bazzocchi, Ignacio A. Jiménez

**Affiliations:** 1Instituto Universitario de Bio-Orgánica Antonio González, Departamento de Química Orgánica, Universidad de La Laguna, Avenida Astrofísico Francisco Sánchez 2, 38206 La Laguna, Tenerife, Spain; 2Instituto de Investigaciones Fármaco Bioquímicas, Facultad de Ciencias Farmacéuticas y Bioquímicas, Universidad Mayor de San Andrés, Avenida Saavedra 2224, Miraflores, La Paz, Bolivia; 3Departamento de Parasitología, Facultad de Farmacia, Universidad Complutense de Madrid, Plaza Ramón y Cajal s/n, 28040 Madrid, Spain; 4Departamento de Farmacia, Bioquímica y Biología Molecular, Universidad CEU-Cardenal Herrera, Avda. Seminario s/n, 46113 Moncada, Valencia, Spain

**Keywords:** *Piper* species, flavonoids, *Giardia intestinalis*, structure-activity relationship, drug-likeness

## Abstract

Diarrhea diseases caused by the intestinal protozoan parasite *Giardia intestinalis* are a major global health burden. Moreover, there is an ongoing need for novel anti-*Giardia* drugs due to drawbacks with currently available treatments. This paper reports on the isolation and structural elucidation of six new flavonoids (**1**–**6**), along with twenty-three known ones (**7**–2**9**) from the *Piper* species. Their structures were established by spectroscopic and spectrometric techniques. Flavonoids were tested for in vitro antiprotozoal activity against *Giardia intestinalis* trophozoites. In addition, structure-activity relationship (SAR) and in silico ADME studies were performed to understand the pharmacophore and pharmacokinetic properties of these natural compounds. Eight flavonoids from this series exhibited remarkable activity in the micromolar range. Moreover, compound **4** was identified as having a 40-fold greater antiparasitic effect (IC_50_ 61.0 nM) than the clinical reference drug, metronidazole (IC_50_ 2.5 µM). This antiprotozoal potency was coupled with an excellent selectivity index (SI 233) on murine macrophages and in silico drug-likeness. SAR studies revealed that the substitution patterns, type of functional group, and flavonoid skeleton played an essential role in the activity. These findings highlight flavonoid **4** as a promising candidate to develop new drugs for the treatment of *Giardia* infections.

## 1. Introduction

Giardiasis, a parasitic disease caused by the flagellate protozoan of *Giardia intestinalis* (also known as *Giardia duodenalis* or *Giardia lamblia*), is a major gastrointestinal parasite, causing an estimated 280 million infections annually [1]. Giardiasis is a neglected tropical disease (NTD) due to the lack of a vaccine, the recurring nature of infections, and the lack of support for new drug discovery. Furthermore, there are limited treatment options and *G. intestinalis* is showing increasing drug resistance [2]. Metronidazole is the recommended first-line treatment for giardiasis, however, several side effects associated with its use have been reported, including risk of carcinogenesis and mutagenesis [3]. Therefore, there is an increasing need for new selective drugs to treat this disease and improve the health of millions of people around the world. In this context, natural products have proved to be an essential source of lead compounds for drug design [4]. Specifically, flavonoids are the richest subclass of phenolic plant constituents in terms of structural and biological diversity [5]. Furthermore, they have become increasingly popular in health protection due to their remarkable biochemical and pharmacological activity including neuroprotective [6], cardioprotective [7], and gastrointestinal protective [8] effects. In addition, flavonoids show antimicrobial activity, such as antibacterial [9], antiviral [10], and antiprotozoal, including trypanocidal [11] and giardicidal effects [12,13]. 

*Piper* is the second largest genus in the Piperaceae family with about 2000 species spread widely over the tropical and subtropical regions of the world [14]. This genus also has a long history in traditional medicine, agriculture, and in the food industry in Central and East Asia, and South and Central America. The therapeutic potential of the *Piper* species has been mainly attributed to amides and flavonoids, and they are well-known for their medicinal contributions to human health [15]. Previous phytochemical studies on *Piper delineatum* [16] have reported the isolation of flavonoids, whereas essential oils, amides, and benzoic acid derivatives are the main metabolites reported from *Piper divaricatum* [17,18] and *Piper glabratum* [19,20].

In ongoing research into natural product chemistry aimed at the discovery of new anti-infective agents against *Giardia* infections, this work reports the isolation, structure elucidation, and evaluation of *Giardia intestinalis* trophozoites of twenty-nine flavonoids (**1**–**29**) from *P. delineatum*, *P. divaricatum*, and *P. glabratum*. Their structures were determined by spectroscopic and spectrometric techniques, including 1D and 2D NMR experiments, and comparison with those reported in the literature. Structure-activity relationship (SAR) and in silico prediction of drug-likeness were also performed.

## 2. Results and Discussion

### 2.1. Chemistry

A bioassay-guided fractionation based on the giardicidal activity of the EtOH extract from *Piper delineatum* leaves, combined with the phytochemical study of *Piper glabratum* and *Piper divaricatum*, yielded twenty-nine flavonoids (**1**–**29**) (Figure 1). The structures of the new compounds **1**–**6** were resolved as described below. 

Compound **1** was isolated as a yellow amorphous solid, and displayed a molecular formula of C_16_H_14_O_5_ by HREIMS analysis. The IR absorption bands at 3376, 1730, 1593, and 759 cm^−1^ showed hydroxyl, carbonyl, and aromatic groups, and the UV absorption maximum at 348 nm was consistent with a chalcone-type derivative [21,22]. The ^1^H NMR spectrum (Table 1) displayed signals at *δ*_H_ 9.04 (2H) and 14.17 (1H), suggesting the presence of three phenolic protons in the molecule. Moreover, the presence of a *trans*-*o*-coumaroyl [(2*E*)-3-(3-hydroxyphenyl)-2-propenoyl] moiety indicated a *meta*-substituted on the basis of the coupling patterns of the aromatic protons [*δ*_H_ 6.91 (d, *J* = 7.6 Hz, H-4), 7.19 (s, H-2), 7.20 (d, *J* = 7.6 Hz, H-6), and 7.28 (t, *J* = 7.6 Hz, H-5)], whereas the large coupling constant (15.6 Hz) between Hα and Hβ signals (*δ*_H_ 7.96 and 7.68) proposed an *E*-geometry of the Cα-Cβ double bond. The substructure of ring A was identified as a tetra-substituted aromatic system in accordance with an AB spin system [*δ*_H_ 6.08 (d, *J*_AB_= 1.6 Hz, H-5′) and 6.00 (d, *J*_AB_ = 1.6 Hz, H-3′)]. In addition, a signal assignable to a methoxy group at *δ*_H_ 3.99 was observed. The ^1^H-^1^H COSY correlations between H-2/H-4, H-4/H-5, H-5/H-6, H-3′/H-5′, and Hα/Hβ confirmed the connectivity deduced from the coupling constant analysis.

The ^13^C NMR spectrum showed peaks corresponding to 16 carbons (Table 1), which were classified into one CH_3_, eight sp2 CH, six sp2 quaternary carbons, and one ketone carbon by analysis from an HSQC experiment. All these data are consistent with a chalcone skeleton with three hydroxy and one methoxy group on tetra-substituted and di-substituted aromatic rings. 

The regiosubstitution of **1** was confirmed by an HMBC experiment, showing a cross-peak between the signal at *δ*_H_ 6.13 (H-5′) with C-1′, C-3′, C-4, and C-6′, and correlation of the signal at *δ*_H_ 6.00 (H-3′) with C-1′, C-2′, C-4′, and C-5′. Long-range ^1^H-^13^C correlation of the phenolic proton at *δ*_H_ 14.17 (OH-2′) to C-1′, C-3′, and C-2′, and those of the signal at *δ*_H_ 3.99 (OCH_3_-6′) to C-6′ (*δ*_C_ 164.4) located the hydroxy and methoxy groups on ring A. The substitution pattern of ring B was supported by correlations of the aromatic protons at *δ*_H_ 6.91 (H-4), 7.19 (H-2), and 7.28 (H-5) with C-3 (*δ*_C_ 158.7). This evidence and comparison with reported data for related compounds [16,22] established the structure of compound **1** as 3,2′,4′-trihydroxy-6′-methoxychalcone. 

The HREIMS of compound **2** provided a [M]^+^ at *m/z* 316.0935, consistent with a molecular formula of C_17_H_16_O_6_. The ^1^H NMR spectrum (Table 1) showed typical chemical shifts and coupling patterns of chalcone-type flavonoids, with a vinylic *trans* system, one chelated hydroxyl proton signal at *δ*_H_ 14.39 (HO-2′), two phenolic proton signals at *δ*_H_ 8.76 (HO-3) and 9.20 (HO-4′), and two methoxy groups at *δ*_H_ 3.76 (CH_3_O-3′) and 3.97 (CH_3_O-6′). These data were sustained by the ^13^C NMR spectrum (Table 1), showing 17 carbon signals, in agreement with the molecular formula. Compound **2** was linked to **1**, showing the most significant differences in the presence of an extra methoxy group (*δ*_H_ 3.76) and the concomitant absence of an aromatic proton signal (*δ*_H_ 6.00 in **1**). Even so, a complete set of 2D NMR spectra was acquired for **2** in order to obtain the complete and unambiguous assignment of the ^1^H and ^13^C NMR chemical shifts and regiosubstitution patterns. Thus, the structure of compound **2** was established as 3,2′,4′-trihydroxy-3′,6′-dimethoxychalcone. Compound **3** gave a molecular formula of C_17_H_16_O_6_ as determined by HREIMS, 30 mass units more than that of compound **1**, indicating the presence of an extra methoxy group compared to **1**. The ^1^H and ^13^C NMR data (Table 1) indicated as the principal differences the replacement of the aromatic methine at C-5 by a methoxy group in **3**, as suggested by the ABC spin system [(δ_H_ 6.49 (H-4), 6.77 (H-6), and 6.81 (H-2)] characteristic of a 1,3,5-trisubstituted aromatic B-ring. These data were confirmed by 2D NMR experiments, establishing the structure of compound **3** as 3,2′,4′-trihydroxy-5,6′-dimethoxychalcone. Compound **4** was closely related to **3,** and its HREIMS (C_17_H_16_O_6_) indicated 14 mass units higher than that of compound **3**, suggesting the presence of an additional methyl group. The great resemblance of their 1D NMR data (Table 1) confirmed the replacement of the OH-4′ signal by one corresponding to a methoxy group (δ_H_ 3.88, δ_C_ 56.1). The full assignment of ^1^H and ^13^C NMR chemical shifts was resolved by 2D NMR experiments, defining the structure of **4** as 3,2′-dihydroxy-5,4′,6′-trimethoxychalcone.

Compound **5** was isolated as a yellow amorphous solid, and its HREIMS spectrum presented a molecular formula of C_18_H_18_O_6_, which was sustained by ^13^C NMR analysis. The IR absorption bands at 3325, 1726, 1608, and 757 cm^−1^ indicated the presence in the molecule of hydroxyl, carbonyl, and aromatic groups, and the absorption band at 283 nm in the UV spectrum suggested the presence of a flavanone skeleton [23]. The ^13^C NMR spectrum (Table 2) exhibited peaks consistent with 18 carbons, which were arranged into three CH_3_, one *sp*^3^ CH_2_, one *sp*^3^ CH, five *sp*^2^ CH, seven *sp*^2^ quaternary carbons, and one keto-carbonyl carbon by an HSQC experiment analysis. The ^1^H NMR spectrum (Table 2) showed 18 proton signals, including a singlet at *δ*_H_ 5.66, characteristic of a phenolic proton, and signals assignable to three methoxy groups at *δ*_H_ 3.79, 3.82, and 3.88. 

The flavanone structure suggested for compound **5** was evidenced by an ABX *spin* [*δ*_H_ 5.30 (*J*_AX_ = 3.0, *J*_BX_ =13.0 Hz, H-2), *δ*_H_ 2.96 (*J*_BX_ = 13.0 Hz, *J*_AB_ = 16.6 Hz, H-3_ax_), and *δ*_H_ 2.78 (*J*_AX_ = 3.0 Hz, *J*_AB_ = 16.6 Hz, H-3_eq_)], an AB *spin* [*δ*_H_ 6.08 (*J*_AB_ = 2.3 Hz, H-6) and *δ*_H_ 6.15 (*J*_AB_ = 2.3 Hz, H-8)], and an ABC *spin* [*δ*_H_ 6.56 (H-6′), 6.53 (H-2′) and 6.40 (*J* = 2.2 Hz, H-4′)] systems, whose multiplicity and coupling constants suggested a tetra-substituted A-ring and a 1,3,5-substituted B-ring on the flavanone core. The structure proposed was confirmed by analysis of 2D NMR data, including HSQC and HMBC experiments. Thus, the methoxy groups were attached at C-5′, C-5, and C-7 based on the ^1^H-^13^C long-range correlations of signals at *δ*_H_ 3.79, 3.88, and 3.82 to carbon signals at δ_C_ 161.4, 162.5, and 166.3, respectively. The absolute configuration at C-2 was proposed as *S* by the levorotatory optical rotation value (−5.2°), the *trans* diaxial *J*_2,3_ of 13.0 Hz [24], and biogenetic means [25]. This evidence and comparison with reported data for related compounds [20] established the structure of **5** as (-)-2*S*-3′-hydroxy-5,7,5′-trimethoxyflavonone.

Compound **6** was closely related to **5** and its HREIMS exhibited a molecular ion at *m/z* 316.1088 (C_17_H_16_O_6_), which was 14 mass units less than the parent compound **5**, suggesting the loss of a methyl group. Its ^1^H NMR spectrum (Table 2) disclosed chemical shifts and coupling pattern of a pyranone unit, typical in flavanones, and the coupling pattern of protons H-2′, H-5′, and H-6′ indicated a 1,3,4-trisubstituted aryl at C-2. 2D NMR experiments let the complete and full assignment of the ^1^H and ^13^C NMR chemical shifts (Table 2), regiosubstitution, and relative configuration of compound **6**. The absolute configuration at C-2 of the flavonoid skeleton was proposed as *S* by the negative optical rotation value (−1.8°) and biogenetic means [24,25]. Thus, the structure for **6** was deduced as (-)-2*S*-3′,4′-dihydroxy-5,7-dimethoxyflavonone.

The known compounds **7**–2**9** were identified by comparison of their spectroscopic data with values reported in the literature as: 2′,4′-dihydroxy-3,6′-dimethoxychalchone (**7**) [22], 2′,4′-dihydroxy-6′-methoxychalchone (**8**) [26], 2′-hydroxy-3,4′,6′-trimethoxychalchone (**9**) [27], 2′,4′-dihydroxy-3,3′,6′-trimethoxychalcone (**10**) [16], 2′,3′,5-trihydroxy-4′,6′,3-trimethoxychalcone (**11**) [16], 2′,3′-dihydroxy-4′,6′-dimethoxychalchone (**12**) [28], 2′,4′,3,4-tetrahydroxy-6′-methoxychalchone (**13**) [29], 2′,4′,4-trihydroxy-3,6′-dimethoxychalchone (**14**) [30], 2′,3,4-trihydroxy-4′,6′-dimethoxychalchone (**15**) [31], uvangoletin (**16**) [32], 7,5′-dihydroxy-5,3′-dimethoxyflavanone (**17**) [16], pinocembrin (**18**) [26], alpinetin (**19**) [26], 7,4′-dihydroxy-5,3′-dimethoxyflavanone (**20**) [33], 8-hydroxy-5,7,3′-trimethoxyflavanone (**21**) [16], 8-hydroxy-5,7-dimethoxyflavanone (**22**) [34], sakuranetin (**23**) [35], 5,7,3′-trimethoxyflavanone (**24**) [36], crysin (**25**) [37], genkwanin (**26**) [38], 3’-hydroxygenkwanin (**27**) [39], galagin (**28**) [37], and isokaemferide (**29**) [40].

### 2.2. Bioassay-Guided Fractionation

The ethanolic (EtOH) extract of *Piper delineatum* leaves was evaluated in vitro against the *Giardia intestinalis* trophozoite stage. Cytotoxicity on the murine macrophage cell line J774 was also measured to check for selectivity. In addition, metronidazole was evaluated for comparative purposes as a reference drug. The EtOH extract showed remarkable giardicidal activity with an IC_50_ (concentration able to inhibit 50% of trophozoites) value of 1.9 μg/mL, and a cytotoxicity (CC_50_, lowest concentration of compound leading to a 50% reduction in cell viability) on murine macrophages of 7.1 μg/mL. This promising result prompted further analysis using a bioassay-guided fractionation (Figure 2). Thus, the EtOH extract was successively partitioned into dichloromethane (DCM), ethyl acetate (EtOAc), and water (H_2_O) fractions, which were assayed against the parasite strain (Table 3). The results highlighted the two organic fractions, DCM (IC_50_ 16.0 μg/mL) and EtOAc (IC_50_ 19.7 μg/mL), as the most promising to continue with fractionation. The active DCM fraction was chromatographed on a silica gel column, and the sub-fractions were combined based on their thin layer chromatographic (TLC) analyses, affording fractions D1 to D10. D3-D8 fractions exhibiting IC_50_ values ranging from 3.3 μg/mL to 9.3 μg/mL were submitted to purification steps by column chromatography to yield flavonoids **1**–**20** (Figure 2, Table 4). Following the same procedure, the active EtOAc fraction was chromatographed to afford sub-fractions E1-E6. Sub-fractions E2 and E3 exhibited IC_50_ values of 4.5 and 3.7 μg/mL, respectively, and therefore, these two sub-fractions were further purified to yield four flavonoids (**21**–**24**). 

### 2.3. Anti-Giardia Activity Assays of Pure Flavonoids

The isolated flavonoids **1**–**24** were tested in vitro for their antiparasitic activity against *Giardia intestinalis* trophozoites. Additionally, the isolated compounds from the inactive sub-fractions (D9 and D10) from *P. delineatum*, and those isolated from *P. divaricatum* and *P. glabratum* (**25**–**29**) were included in the present study in order to broaden the SAR and pharmacophore model studies. To test the safety profile of the evaluated compounds, they were screened for their cytotoxic effect on the murine macrophage cell line J774. 

Previous studies on the giardicidal activity of the known isolated compounds **18** (pinocembrine) and **25** (chrysin) [41], and those of compound **28** (galangin) [42] have been reported. Nevertheless, these compounds have been included to broaden the SAR analysis, for comparative purposes, and due to the strain and procedures being different from the ones used herein. The results of the anti-*Giardia* activity (Table 4) revealed that compound **4**, having a chalcone skeleton with a tetrasubstituted A-ring and a trisubstituted B-ring, exhibited potent activity with an IC_50_ value of 0.061 μM, 41-fold more potent than metronidazole (IC_50_ 2.5 μM), used as the reference drug. This potency was combined with low cytotoxicity on murine macrophages, showing a CC_50_ value of 14.2, and an excellent selectivity index (SI) of 233. Furthermore, results also indicated that flavonoids **1**, **3**, **7**–**9**, **12,** and **14** showed potent activity with IC_50_ values ranging from 4.7 to 10.8 μM, slightly lower than the reference drug. Moreover, compounds **3**, **7**, **8,** and **9** showed some degree of selectivity on the murine macrophages with a SI ≥ 2. Compounds **2**, **10**, **11**, **16**, **23,** and **24** showed weak activity (IC_50_ 22.5–70.5 μM), whereas the remaining assayed compounds were inactive (IC_50_ > 100 μM). In this study, the identification of a potent and selective flavonoid as a promising lead compound supports the effectiveness of the strategy of bioassay-guided fractionation and isolation, as a chemo-selective purification methodology to access bioactive natural products.

### 2.4. Structure-Activity Relationship Analysis 

To our knowledge, this is the first study on the structure-giardicidal relationship of natural chalcones. Moreover, there has only been one report on cyclic analogue flavonoids [12]. Therefore, the influence of the substitution pattern of the tested flavonoids on their giardicidal activity has been analysed considering three regions of the 1,3-diarylpropane skeleton (C_6_-C_3_-C_6_): the propane unit and the regiosubstitution pattern of the two aromatic moieties (A and B rings). This analysis has revealed the following features of the structure-activity relationship (SAR) (Table 4, Figure 3). (a) The propane unit seems essential for the activity. Thus, the presence of an α,β-unsaturated carbonyl system strongly affects the activity, since a double bond reduction leads to a decrease in the effect (chalcone **8** *versus* dihydrochalcone **16**). Furthermore, cyclization in flavonoids causes a loss of activity as revealed by the comparison of the potency between open chalcones with their counterpart cyclic analogues (**3** vs. **17**, **4** vs. **5**, **8** vs. **19**, **12** vs. **22,** and **14** vs. **20**). Thus, the results showed that the most effective motif for this fragment was a chalcone scaffold, an open chain flavonoid in which the two aromatic rings are joined by a three-carbon α,β-unsaturated carbonyl system. (b) The effect of substitution in the B-ring suggests that a trisubstituted phenyl ring, in which a hydroxyl at C-3 and a methoxyl at C-5 seem crucial for the activity. Thus, the replacement of a phenol group by hydrogen caused a dramatic loss in potency (**4** vs. **9**). Moreover, the substitution pattern of the B ring plays a modulator effect on the activity as revealed by comparison in the potency of flavonoids **1** vs. **8**, **2** vs. **10**, **7** vs. **8**, **11** vs. **12**, and **13** vs. **14**. Furthermore, a 3,4-*orto*-substituted B-ring clearly confers greater activity properties, since hydrogen replacement by a hydroxyl group triggers partial (**7** vs. **14**) or total loss (**1** vs. **13**) of activity. (c) Regarding the effect of substitution in the A-ring, a tetrasubstituted phenyl ring, a hydroxyl at C-2′, and methoxyl groups at C-4′ and C-6′ appear to be important for activity against *G. instestinalis*. In addition, the replacement of a 4′-OH-benzene ring by a methoxyl strongly affects potency (**4** vs. **3**). Furthermore, the hydrogen atom at the C-3′ position also plays a significant role in activity, as its replacement affects the bioactivity profile, leading to a moderate or dramatic loss of antiparasitic activity (**1** vs. **2**, **7** vs. **10**, and **4** vs. **11**).

In general, the overall oxidation level of the molecule, the substitution patterns, and the type of functional group are important features of this series of flavonoids for anti-*Giardia* activity. In particular, hydroxyl and methoxyl substituents at *meta*-position on phenyl rings combined with a conjugated enone motif may play a key role in the binding interactions with target molecules modulating their giardicidal potency. These results are in agreement with previous reports [43,44] and expand the chemical space of flavonoids for anti-giardia drug discovery. 

### 2.5. In silico Drug-Likeness Predictions

Introducing a new drug on the market is a risky, complex, time-consuming, and expensive process, in which the physicochemical and pharmacokinetic properties play a crucial role since they can affect the pharmacological profile and the performance of a drug candidate. In fact, about 96% overall unsuccessful rate in drug development is due to undesirable pharmacokinetic profiles [45]. Nowadays, to overcome this negative outlook, the prediction of pharmacokinetic profiles such as drug absorption, distribution, metabolism, excretion, and toxicity (ADMET) is widely accepted as an important tool in drug design and lead optimization to anticipate possible clinical suitability [46]. In order to better understand the overall properties of the reported flavonoids and increase the success percentage of these compounds going up to further stages of development, the QikProp module of Schrödinger software [47] was used as a mathematical method for predicting in vivo physicochemical and pharmacokinetic properties. These parameters furnish insights into key aspects such as drug-likeness, permeability, solubility, oral absorption, bioavailability, or metabolism (Appendix A and Table 5).

The first parameter that was considered was #stars, which informs about the number of properties of each flavonoid that fail to remain within the standard ranges, therefore, a lower number of #stars appears for a better druglike molecule [47]. Chemo-informatic analysis exhibited that all the analyzed flavonoids fulfilled this descriptor, showing the best possible result for this descriptor (#stars = 0). Lipophilicity (QP log Po/w) is another crucial parameter requirement for a potential drug and plays a remarkable role in absorption, bioavailability, hydrophobic interactions with the drug, metabolism, excretion, cytotoxicity, and in vivo pharmacological properties. The predicted values of analysed compounds were within the permissible range of this parameter requirement [48]. Moreover, the predicted values for brain/blood partition coefficient (QP log BB), and those of QP PMDCK and QP PCaco, a good mimic for kidney and epithelial permeability, were found to be great for selected flavonoids. In addition, the percentage of human oral absorption (% HOA) was found to be similar to or higher than 80% for selected flavonoids, indicating they possess suitable passive oral absorption. The metabolic reactions descriptor (#metab) shows how efficiently a drug is transformed into metabolites after being absorbed, distributed, and excreted, and all the compounds were within the recommended range. Distribution is pending on various parameters, such as the quantity of binding to plasma proteins (QP log Khsa), and permeability of the central nervous system (QP log S). In these parameters, all compounds had a great profile in concordance with standard values. Properties of chemical moieties, such as molecular weight, hydrogen bond donor (HBD), and hydrogen bond acceptor (HBA), play a crucial role to modulate the drug-likeness of molecules. All of them fully complied with the recommended ranges. Moreover, solvent-accessible surface area (SASA), which can be used as a size-related parameter that affects the partition coefficient and aqueous solubility, gave satisfactory values [45]. Moreover, none of these flavonoids violated Lipinski’s rule of five, thus interpreting that these compounds possess outstanding pharmacokinetic properties. Overall, this analysis reveals that the most potent compound, 2′,3-dihydroxy-4′,6′,5-trimethoxychalcone (**4**), exhibits exceptional drug-likeness since it meets all the pharmacokinetic criteria [48]. Even so, these results are not in concordance with previous reports in which the pharmacokinetic properties of flavonoids were unsatisfactory [49]. Therefore, the bioavailability of flavonoids depends upon their basic skeleton, substitution patterns, and functional group. The above results provided powerful information for the further development of flavonoids as anti-giardia agents.

## 3. Materials and Methods

### 3.1. General 

Optical rotations were recorded on a Perkin Elmer 241 automatic polarimeter in CHCl_3_ at 25 °C and the [α]_D_ values are given in 10^−1^ deg cm^2^/g. UV spectra were measured in absolute EtOH on a JASCO V-560 spectrophotometer. IR (film) spectra were measured on a Bruker IFS 55 spectrophotometer. The NMR experiments were recorded on Bruker Avance 400 and 500 spectrometers with the pulse sequences given by Bruker using (CD_3_)_2_CO, CDCl_3_, or C_6_D_6_ as solvents, and the chemical shifts are given in δ (ppm) with TMS as an internal reference. EIMS and HREIMS were performed with a Micromass Autospec spectrometer. Silica gel 60 (particle size 15–40 and 63–200 μm, Macherey-Nagel) and Sephadex LH-20 (Pharmacia Biotech, Barcelona, Spain) were used for column chromatography, silica gel 60 F254 (Macherey-Nagel) was used for analytical or preparative TLC, and nanosilica gel 60 F254 (Macherey-Nagel) for high-performance TLC (HPTLC). Centrifugal preparative TLC (CPTLC) was made using a Chromatotron (Harrison Research Inc. model 7924T) on 1, 2-, or 4-mm silica gel PF254 disks with a flow rate of 2–4 mL/min. The spots were followed by UV light and heating silica gel plates sprayed with H_2_O-H_2_SO_4_-AcOH (1:4:20). All solvents used were of analytical grade and purchased from Panreac (Barcelona, Spain). For bioassays, TYI-S-33, fetal calf serum, bovine bile, fetal bovine serum, RPMI-1640, penicillin G Penilevel^®^ 100.000 U.I. (ERN laboratories, Barcelona, Spain), and streptomycin sulphate (Normon, Madrid, Spain) were used. Metronidazole, used as a reference drug, was purchased from Sigma-Aldrich, St Louis, MO, USA.

### 3.2. Plant Material 

Leaves of *Piper delineatum* Trel. and leaves of *Piper divaricatum* G. Mey. were harvested in Iquitos, Maynas Province, Department of Loreto, Perú (November 2009). A voucher specimen (10484) and (10548), respectively, were identified by Juan Ruiz Macedo and placed in the Amazonense Herbarium of the Universidad Nacional de la Amazonia Peruana, Iquitos, Peru. Fruits of *Piper glabratum* (Kunth) Steud. were collected by Dra. Genevieve Bourdy in Alto Beni, Abel Iturralde Province, Department of La Paz, Bolivia, in September 2002. A voucher specimen (GB-1877), identified by botanist Ricardo Callejas-Posada (Universidad de Antioquia, Medellin, Colombia) was deposited in the National of La Paz Herbarium, Bolivia.

### 3.3. Extraction, Bioassay-Guided Fractionation, and Isolation

The air-dried and powdered leaves of *P. delineatum* (252.2 g) were extracted in a Soxhlet apparatus with 4 L of 96% ethanol (EtOH) until exhaustion. The solvent was removed and rated to afford 57.3 g of extract, which was assayed on *G. intestinalis* trophozoites giving a potent activity. Therefore, the ethanolic extract was suspended in water and solvent-solvent partitioned sequentially with dichloromethane (DCM) and ethyl acetate (EtOAc). The organic phases were concentrated under reduced pressure to give DCM (30.0 g) and EtOAc (3.3 g) fractions, whereas the aqueous residue (H_2_O) was lyophilized providing the H_2_O fraction (20.5 g). Biological evaluation revealed that the organic fractions were active against the strain of *Giardia intestinalis* and were further investigated (Figure 2).

The DCM (30 g) residue was subjected to a silica gel column chromatography, using mixtures of hexanes/EtOAc of increasing polarity (10:0 to 0:10) as eluent to yield nineteen sub-fractions, which were combined based on their TLC profile (D1-D10). Giardicidal activity revealed that sub-fractions D3-D8 were active, and were subjected to several chromatography steps until obtaining the pure compounds. Each sub-fraction was fractioned by size exclusion chromatography on Sephadex LH-20 and eluted with isocratic mixtures of hexanes/CHCl_3_/MeOH (2:1:1) or CHCl_3_/MeOH (1:1), affording around thirty sub-fractions, which were combined on the basis of their TLC profiles. Then, each of these sub-fractions was subjected to centrifugal preparative thin layer chromatography (CPTLC) on silica gel and eluted with different mixtures of increasing polarity, affording around fifty sub-fractions, which were combined on the basis of their TLC profiles and purified by preparative TLC to give the pure compounds (**1**–**5** and **7**–**24**), as described in detail in the Appendix A. Based on the bioguided-assay results in which the antiparasitic activity seems to be related to the presence of flavonoids, and in order to complete the phytochemical study, the inactive sub-fractions (D1, D2, D9, and D10) were analyzed by ^1^H NMR and TLC. The results revealed the presence of flavonoids in sub-fractions D9 and D10, which were further investigated, affording compounds **17**, **20,** and **21** (Appendix A). The active EtOAc fraction (3.3 g) was subjected to a silica gel column chromatography, using mixtures of CH_2_Cl_2_/EtOAc of growing polarity (200 mL of 8:2, 6:4, 4:6, 2:8, and 0:10) and finally MeOH as eluent to afford sub-fractions E1-E6. Preliminary giardicidal activity analysis revealed that E2 and E3 sub-fractions were active against the strain of *G. intestinalis*, which were further investigated to afford compounds **1**, **3**, **7**, and **17** (Appendix A).

The air-dried and powdered leaves of *P. divaricatum* (343.2 g) were extracted in a Soxhlet apparatus with 96% ethanol (4 × 4 L) until exhaustion. The solvent was removed to afford 75.7 g of crude extract. The ethanolic extract (EtOH) was suspended in water and solvent-solvent partitioned sequentially with dichloromethane (DCM) and ethyl acetate (EtOAc). The resulting organic phases were concentrated under reduced pressure to give DCM (29.8 g) and EtOAc (0.6 g) fractions, whereas the aqueous residue (H_2_O) was lyophilized providing the H_2_O fraction (12.1 g). The CH_2_Cl_2_ fraction was subjected to a silica gel column chromatography, using mixtures of hexanes/EtOAc of increasing polarity (from 10:0 to 0:10, 2 L) as eluent to afford seven fractions (A-G). Preliminary ^1^H NMR and TLC analysis revealed the presence of flavonoids in fractions D (hexanes/EtOAc, 7:3), F (hexanes/EtOAc, 3:7), and G (hexanes/EtOAc, 0:10) which were further investigated to give compounds **6**, **15**, **23**, **26**, **27,** and **29** (Appendix A).

The air-dried fruits of *P. glabratum* (15.9 g) were crushed and extracted with EtOH 70% (2 × 400 mL) by maceration. Evaporation of the solvent under reduced pressure provided 3.2 g of crude extract, which was successively partitioned into CH_2_Cl_2_/H_2_O (1:1, *v*/*v*) solution. Removal of the CH_2_Cl_2_ from the organic-soluble extract under reduced pressure yielded 2.1 g of residue, whereas the aqueous residue (H_2_O) was lyophilized providing the H_2_O fraction (0.5 g). The CH_2_Cl_2_ residue was subjected to a silica gel column chromatography, using mixtures of hexanes/EtOAc of growing polarity (from 9:1 to 0:10, 200 mL,) as eluent to afford nine fractions (A-I). Preliminary ^1^H NMR and TLC analysis revealed the presence of flavonoids in C and E sub-fractions, which were further chromatographed to give compounds **18**, **28,** and **25** (Appendix A).

#### 3.3.1. Compound (1). 2′,4′,3-Trihydroxy-6′-methoxychalcone 

Yellow amorphous solid; UV λ_max_ (MeOH) (log ε) 206 (5.1), 348 (4.9) nm; IR ν_max_ (film) 3376, 2957, 2853, 1730, 1593, 1462, 1331, 1262, 1205, 1163, 1110, 1029, 802, 759, 712 cm^−1^; ^1^H and ^13^C NMR data, see Table 1; EIMS *m/z* 286 [M^+^] (98), 269 (12), 258 (9), 193 (100), 167 (44), 69 (19); HREIMS *m/z* 286.0875 [M^+^] (calcd for C_16_H_14_O_5_, 286.0841). 

#### 3.3.2. Compound (2). 2′,4′,3-Trihydroxy-3′,6′-dimethoxychalcone

Yellow amorphous solid; UV λ_max_ (MeOH) (log ε) 207 (4.3), 351 (4.0) nm; IR ν_max_ (film) 3358, 2922, 2852, 1726, 1628, 1586, 1450, 1329, 1276, 1196, 1109, 1010, 977, 858, 792, 736 cm^−1^; ^1^H and ^13^C NMR data, see Table 1; EIMS *m/z* 316 [M^+^] (100), 230 (8), 223 (19), 196 (49), 181 (32), 167 (11), 153 (19), 139 (7), 91 (6); HREIMS *m/z* 316.0935 [M^+^] (calcd for C_17_H_16_O_6_, 316.0947). 

#### 3.3.3. Compound (3). 2′,4′,3-Trihydroxy-6′,5-dimethoxychalcone

Yellow amorphous solid; UV λ_max_ (MeOH) (log ε) 210 (5.0), 348 (4.7) nm; IR ν_max_ (film) 3408, 2955, 2853, 1729, 1660, 1592, 1531, 1463, 1279, 1196, 1114, 1066, 957, 836, 730 cm^−1^; ^1^H and ^13^C NMR data, see Table 1; EIMS *m/z* 316 [M^+^] (81), 257 (8), 193 (100), 167 (50), 69 (16); HREIMS *m/z* 316.0967 [M^+^] (calcd for C_17_H_16_O_6_, 316.0947).

#### 3.3.4. Compound (4). 2′,3-Dihydroxy-4′,6′,5-trimethoxychalcone

Yellow amorphous solid; UV λ_max_ (MeOH) (log ε) 211 (4.9), 345 (4.7) nm; IR ν_max_ (film) 3326, 2924, 2851, 1726, 1622, 1582, 1431, 1386, 1284, 1210, 1112, 1049, 935, 855, 791, 758 cm^−1^; ^1^H and ^13^C NMR data, see Table 1; EIMS *m/z* 330 [M^+^] (100), 313 (9), 271 (9), 207 (97), 181 (35), 151 (11), 69 (6); HREIMS *m/z* 330.1090 [M^+^] (calcd for C_17_H_16_O_6_, 330.1103). 

#### 3.3.5. Compound (5). (-)-(2S)-3′-Hydroxy-5,7,5′-trimethoxyflavanone 

Yellow amorphous solid; [α]_D_^20^ -5.2 (*c* 0.13, MeOH); UV λ _max_ (MeOH) (log ε) 203 (4.7), 227 (4.4), 283 (4.2) nm; IR ν_max_ (film) 3325, 2926, 2853, 1726, 1608, 1461, 1364, 1271, 1159, 1113, 1071, 945, 824, 757 cm^−1^; ^1^H and ^13^C NMR data, see Table 2; EIMS *m/z* 330 [M^+^] (100), 300 (10), 207 (79), 193 (12), 181 (53), 151 (10), 137 (11), 120 (10), 57 (9); HREIMS *m/z* 330.1115 [M^+^] (calcd for C_18_H_18_O_6_, 330.1103). 

#### 3.3.6. Compound (6). (-)-(2S)-3′,4′-Dihydroxy-5,7-dimethoxyflavanone 

Yellow amorphous solid; [α]_D_^20^ -1.8 (*c* 0.25, MeOH); UV λ_max_ (MeOH) (log ε) 204 (5.2), 226 (5.0), 283 (4.9) nm; IR ν_max_ (film) 3399, 2943, 1728, 1652, 1607, 1569, 1523, 1457, 1428, 1272, 1216, 1158, 1111, 1038, 959, 870, 820, 753 cm^−1^; ^1^H and ^13^C NMR data, see Table 2; EIMS *m/z* 316 [M^+^] (100), 288 (22), 207 (54), 194 (47), 181 (69), 154 (15), 137 (20), 123 (10); HREIMS *m/z* 316.1088 [M^+^] (calcd for C_18_H_18_O_6_, 316.0947).

### 3.4. Biological Assays 

#### 3.4.1. Parasites 

Trophozoites of *Giardia* (strain ATCC^®^ 203333) were grown in modified TYI-S-33 medium at pH 7.0, supplemented with 10% heat-inactivated fetal calf serum and 0.5 g bovine bile (Sigma-Aldrich, St Louis, MO, USA), in 20 mL screw-cap culture vials.

#### 3.4.2. Cells

J774 murine macrophages (strain ATCC) were grown and maintained in RPMI-1640 medium supplemented with 10% heat-inactivated FBS (Sigma-Aldrich, St Louis, MO, USA), penicillin G (100 U/mL) (Penilevel, Laboratorios ERN, S.A., Barcelona, Spain), and streptomycin (100 μg/mL) (laboratorios Normon, S. A., Madrid, Spain) at 37 °C and 5% CO_2_ air atmosphere.

#### 3.4.3. In Vitro Assay on Trophozoite Form of Giardia Intestinalis

The assay was carried out according to a previously described method [50]. Detachment of trophozoites for preparation of inoculant was achieved by chilling the cultures on ice for 20 min. Trophozoites (5 × 10^4^ parasites/well) were cultured in 96-well plastic plates (Sarstedt, Nümbrecht, Germany). Compounds were dissolved in DMSO. Different dilutions of the tested samples (100, 50, 25, 12.5, 6.25, 3.12, 1.56, and 0.78) up to 200 μL final volume were added. After 48 h incubation at 37 °C, the growth medium was removed, and then 2.5 mM resazurin in PBS-2% glucose was added and the plates returned to the incubator for another 3 h. To evaluate cell viability, the fluorescence intensity (535 nm-excitation wavelength- and 590 nm-emission wavelength) was determined with a fluorimeter Infinite 200 (Tecan i-Control, Tecan Ibérica Instrumentation S.L., Barcelona, Spain) to calculate the growth inhibition rate (%). All tests were carried out in triplicate. Metronidazole was used as a reference drug. The efficacy of each compound was estimated by calculating the IC_50_ (concentration of the compound that produced a 50% reduction in parasites).

#### 3.4.4. Cytotoxicity Assay on Murine Macrophages

The assay was carried out according to a previously described method [51]. The murine macrophage J774 cell line was seeded (5 × 10^4^ cells/well) in 96-well flat-bottom plates with 100 μL of RPMI-1640 medium. The cells were allowed to attach for 2 h at 37 °C, 5% CO_2_, and then 100 μL of RPMI-1640 medium containing the test sample at different concentrations (100, 50, 25, 12.5, 6.25, 3.12, 1.56, and 0.78 µg/mL) was added to the final volume and exposed for another 48 h. Growth controls and signal-to-noise were also included. Afterward, a volume of 20 μL of 2.5 mM resazurin solution in PBS was added, and plates were placed again in the incubator for another 3 h to evaluate cell viability. The reduction of resazurin was determined by fluorometry as in the promastigote assay. Each concentration was assayed in triplicate. The cytotoxicity effect of compounds was defined as the 50% reduction of cell viability of treated culture cells with respect to untreated culture (CC_50_).

#### 3.4.5. Statistical Analysis 

All values were expressed as means ± standard error of the mean (S.E.M.). Statistically significant differences between treated and control samples were tested by one-way analysis of variance (ANOVA). The efficacy against parasites (IC_50_) and cytotoxicity effect (CC_50_) of samples were calculated from Probit analysis, using SPSS v20.0 software. 

## 4. Conclusions

The current study reports on our efforts to discover new drug candidates for giardiasis to overcome several drawbacks of current treatments, such as limited accessibility, drug-related side effects as well as the emergence of multidrug-resistant parasite strains. Therefore, a series of flavonoids isolated from *Piper* species and their evaluation on the *Giardia intestinalis* strain, together with structure-activity relationship and in silico ADME studies, was carried out to find new lead compounds for the treatment of giardiasis. We have successfully identified eight flavonoids from this series exhibiting remarkable activity against *G. intestinalis* trophozoites in the micromolar range, and favourable predicted pharmacokinetic properties. In particular, compound **4**, with a chalcone skeleton, a tetrasubstituted A-ring, and a trisubstituted B-ring displays an inhibitory effect 40-fold higher than the clinically reference drug, metronidazole. This anti-*Giardia* potency is coupled with an excellent selectivity index and favourable drug-likeness. 

In summary, the current study provides insight into the antiprotozoal potential for the clinical application of chalcone-related flavonoids. In fact, the 2′,3-dihydroxy-4′,6′,5-trimethoxychalcone **4** is a hopeful lead compound and deserves further studies to unravel its potential as a therapeutic alternative to the giardiasis current treatment.

## Figures and Tables

**Figure 1 pharmaceuticals-15-01386-f001:**
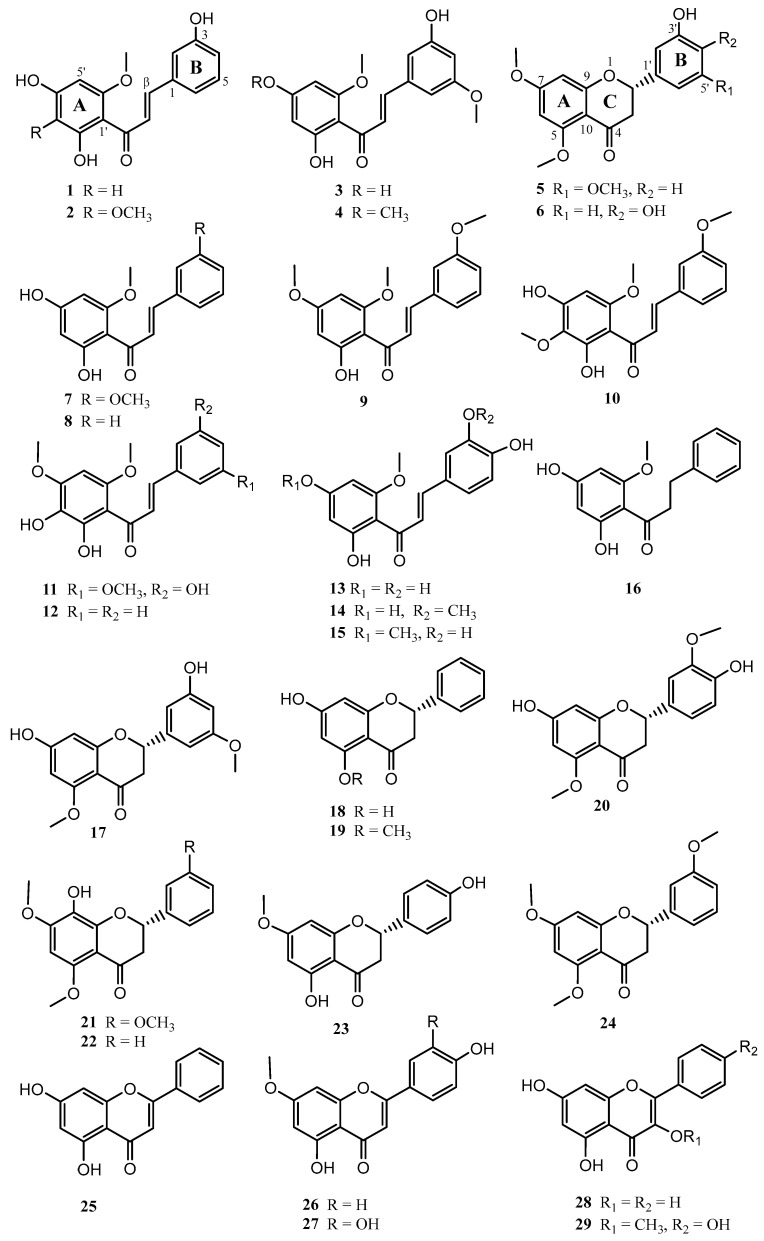
Chemical structure of flavonoids isolated from *P. delineatum*, *P. divaricatum*, and *P. glabratum*.

**Figure 2 pharmaceuticals-15-01386-f002:**
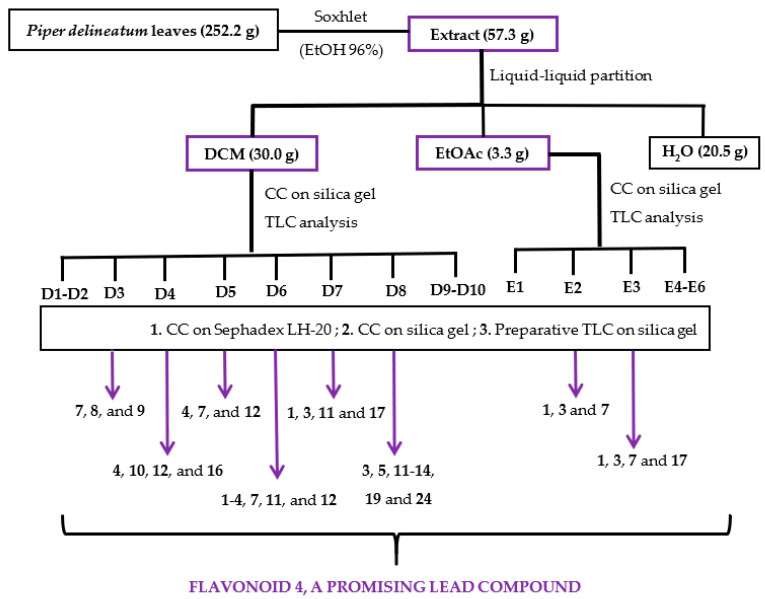
Flowchart of bioguided isolation of *P. delineatum* leaves against *G. intestinalis* trophozoites.

**Figure 3 pharmaceuticals-15-01386-f003:**
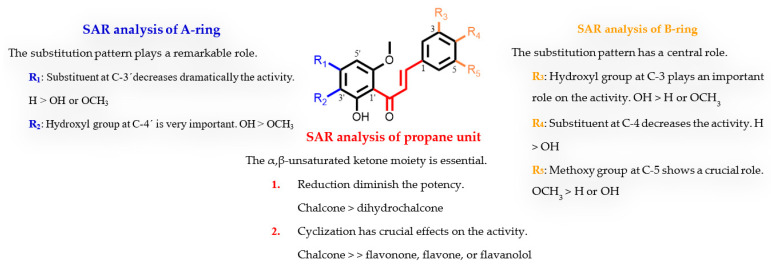
Overview of the structural requirements of flavonoids as potential giardicidal agents.

**Table 1 pharmaceuticals-15-01386-t001:** ^1^H (600 MHz) *^a^* and ^13^C (100 MHz) NMR (δ, (CD_3_)_2_CO) data *^b^* of compounds **1**–**4**.

	1		2		3		4	
	δ_H_	δ_C_	δ_H_	δ_C_	δ_H_	δ_C_	δ_H_	δ_C_
1		137.8 s		137.7 s		138.4 s		138.3 s
2	7.19 s	115.3 d	7.19 s	115.3 d	6.81 s	108.3 d	6.81 s	108.4 d
3		158.7 s		158.8 s		159.8 s		159.8 s
4	6.91 d (7.6)	118.2 d	6.92 d (8.0)	118.4 d	6.49 s	104.3 d	6.50 s	104.4 d
5	7.28 t (7.6)	131.0 d	7.29 t (8.0)	130.9 d		162.2 s		162.3 s
6	7.20 d (7.6)	121.0 d	7.20 d (8.0)	120.9 d	6.77 s	106.2 d	6.78 s	106.3 d
C=O		193.1 s		193.7 s		193.1 s		193.4 s
α	7.96 d (15.6)	128.4 d	7.96 d (15.6)	128.2 d	7.93 d (15.5)	128.6 d	7.93 d (15.7)	128.6 d
β	7.68 d (15.6)	142.8 d	7.69 d (15.6)	143.1 d	7.62 d (15.5)	142.8 d	7.64 d (15.7)	143.2 CH111
1′		106.3 s		106.5 s		106.3 s		106.9 s
2′		169.0 s		160.8 s		169.0 s		169.1 s
3′	6.00 d (1.6)	97.0 d		130.0 s	6.00 s	97.0 d	6.10	94.7 d
4′		166.0 s		158.1 s		166.1 s		167.6 s
5′	6.08 d (1.6)	92.3 d	6.12 s	91.7 d	6.08 s	92.3 d	6.13	91.9 d
6′		164.4 s		159.7 s		164.3 s		163.8 s
OCH_3_-5					3.82 s	55.6 q	3.82 s	55.6 q
OCH_3_-3′			3.76 s	60.6 q				
OCH_3_-4′							3.88 s	56.1 q
OCH_3_-6′	3.99 s	56.5 t	3.97 s	56.5 q	3.98 s	56.4 q	4.01 s	56.6 q
OH-2′	14.17 s		14.39 s		14.16 s		14.16 s	
OH-4′	9.04 s *^c^*		9.20 s		9.10 s *^c^*			
OH-3	9.04 s *^c^*		8.76 s		9.10 s *^c^*		8.60 s	

*^a^ J* are given in parentheses in Hz. *^b^* Data based on COSY, HSQC, and HMBC experiments. *^c^*Overlapping signals.

**Table 2 pharmaceuticals-15-01386-t002:** ^1^H (400 MHz) *^a^* and ^13^C (100 MHz) NMR (δ) data *^b^* of compounds **5** and **6**.

	5 *^c^*	5 *^c^*	6 *^d^*	6 *^e^*	6 *^d^*
	δ_H_	δ_C_	δ_H_	δ_H_	δ_C_
2	5.30 dd (3.0, 13.0)	79.1 CH	5.32 dd (2.9, 12.6)	5.00 dd (2.9, 12.8)	79.9 CH
3	2.78 dd (3.0, 16.6) 2.96 dd (13.0, 16.6)	45.7 CH_2_	2.59 dd (2.9, 16.3) 2.94 dd (12.6, 16.3)	2.59 dd (2.9, 16.3) 2.94 dd (12.8, 16.3)	46.3 CH_2_
4		189.5 C			188.4 C
5		162.5 C			163.2 C
6	6.08 d (2.3)	93.4 CH	6.17 d (2.3)	6.07 d (2.2)	93.5 CH
7		166.3 C			166.6 C
8	6.15 d (2.3)	93.8 CH	6.14 d (2.3)	5.98 d (2.2)	94.4 CH
9		165.1 C			165.8 C
10		106.0 C			106.7 C
1′		141.4 C			132.0 C
2′	6.53 s	105.7 CH	7.01 s	7.16	114.6 CH
3′		157.6 C			146.0 C
4′	6.40 t (2.2)	101.7 CH			146.3 C
5′		161.4 C	6.85 s *^f^*	7.01 d (8.1)	116.0 CH
6′	6.56 s	104.3 CH	6.85 s *^f^*	6.72 dd (1.9, 8.1)	119.1 CH
OMe-5	3.88 s	56.3 CH_3_	3.82 s	3.24 s	56.2 CH_3_
OMe-7	3.82 s	55.8 CH_3_	3.85 s	3.40 s	56.0 CH_3_
OMe-5′	3.79 s	55.6 CH_3_			
OH-3′	5.66 s				

*^a^ J* are given in parentheses in Hz. *^b^* Data based on COSY, HSQC, and HMBC experiments. *^c^* Spectra recorded in CDCl_3_. *^d^* Spectra recorded in (CD_3_)_2_CO. *^e^* Spectra recorded in C_6_D_6_ at 500 MHz. *^f^* Overlapping signal.

**Table 3 pharmaceuticals-15-01386-t003:** Activity against trophozoite stage of *Giardia intestinalis* and cytotoxicity against the murine macrophage J774 cell line of crude extract, fractions, and sub-fractions from *Piper delineatum* leaves.

Extract or Fractions	*G. intestinalis* IC_50_ *^a,b^* (µg/mL)	Macrophages CC_50_ *^c^* (µg/mL)	SI *^d^*
EtOH	1.9 ± 1.5	7.1 ± 0.6	3.7
CH_2_Cl_2_	16.0 ± 1.2	6.6 ± 0.1	0.4
D3	5.6 ± 1.3	15.5 ± 1.0	2.8
D4	3.3 ± 0.7	9.2 ± 1.2	2.8
D5	4.7 ± 0.1	10.9 ± 0.6	2.3
D6	9.3 ± 1.5	17.5 ± 1.2	1.9
D7	5.0 ± 1.1	4.5 ± 0.9	0.9
D8	7.5 ± 0.5	10.6 ± 1.4	1.4
EtOAc	19.7 ± 0.7	4.9 ± 0.4	0.3
E2	3.7 ± 1.3	3.2 ± 0.1	0.9
E3	4.5 ± 0.9	2.7 ± 0.1	0.6
Metronidazole *^e^*	0.4 ± 0.1	233	582

*^a^* IC_50_: concentration able to inhibit 50% of trophozoites. *^b^*Fractions and sub-fractions not included were inactive, IC_50_ > 20 µM. *^c^* CC_50_ concentration able to inhibit 50% of murine macrophages. *^d^* SI: selectivity index (CC_50_/IC_50_). *^e^* Metronidazole was used as a positive control.

**Table 4 pharmaceuticals-15-01386-t004:** Activity against trophozoite stage of *Giardia intestinalis* and cytotoxicity against the murine macrophage J774 cell line of flavonoids **1**–**29** isolated from *Piper* species.

Compound	*G. intestinalis* IC_50_*^a^*^,*b*^ (µM)	Macrophages CC_50_ *^c^* (µM)	SI *^d^*
**1**	7.0 ± 0.1	12.9 ± 0.2	1.8
**2**	47.7 ± 1.2	184.4 ± 2.6	3.9
**3**	6.0 ± 0.3	12.0 ± 0.1	2.0
**4**	0.061 ± 0.001	14.2 ± 0.2	233
**7**	4.7 ± 0.1	15.7 ± 0.2	3.3
**8**	9.6 ± 0.2	37.8 ± 1.1	3.9
**9**	10.8 ± 0.1	22.6 ± 0.5	2.1
**10**	32.7 ± 1.3	44.2 ± 0.7	1.4
**11**	22.5 ± 0.8	48.5 ± 0.8	2.2
**12**	7.3 ± 0.6	25.3 ± 0.4	3.5
**14**	10.4 ± 0.2	16.5 ± 0.03	1.6
**16**	70.5 ± 0.3	>367.2	>5.2
**23**	52.4 ± 2.1	184.2 ± 2.3	3.5
**24**	49.3 ± 0.9	144.5 ± 2.3	2.9
Metronidazole *^e^*	2.5 ± 0.1	>584.3	>233

*^a^* IC_50_: concentration able to inhibit 50% of trophozoites. *^b^* Compounds not included were inactive IC_50_ > 100 µM. ^*c*^ CC_50_ concentration able to inhibit 50% of murine macrophages. *^d^* SI: selectivity index (CC_50_/IC_50_). *^e^* Metronidazole was used as a positive control.

**Table 5 pharmaceuticals-15-01386-t005:** In silico ADME profile prediction of selected flavonoids *^a^* and their range/recommended values *^b^*.

Property	1	3	4	7	8	9	12	14	Range/Values
#stars	0	0	0	0	0	0	0	0	0–5
QPlogBB	−1.744	−1.850	−1.325	−1.246	−1.150	−0.717	−1.129	−1.731	−3.0 to 1.2
QPPCaco	161.650	162.190	533.788	523.193	523.287	1724.836	661.863	161.090	<25 poor, >500 great
QPPMDCK	69.011	69.259	250.997	245.616	245.664	891.757	316.681	68.752	<25 poor, >500 great
QPlogKhsa	−0.022	0.003	0.179	0.172	0.158	0.206	0.175	−0.013	−1.5 to 1.5
QPlogPo/w	2.265	2.375	3.204	3.118	3.017	3.836	3.212	2.324	−2.0 to 6.5
QPlogKp	−3.299	−3.396	−2.441	−2.354	−2.255	−1.398	−2.111	−3.367	−8.0 to −1.0
QPlogS	−3.555	−3.815	−4.242	−4.115	−3.869	−4.535	−4.072	−3.401	−6.5 to 0.5
#metab	4	5	5	4	3	4	4	5	1 to 8
%HOA	79.738	80.409	94.518	93.864	93.271	100	96.241	80.055	>80% high, <25% poor
PSA	95.0	103.3	89.0	80.8	72.5	66.5	78.0	102.4	7.0 to 200.0
SASA	555.1	592.0	616.0	580.9	543.4	604.5	578.5	567.6	300.0 to 1000.0
Mol MW	286.28	316.31	330.34	300.31	270.28	314.34	300.31	316.31	130.0 to 725.0
#rotor	8	9	9	8	7	8	8	9	0 to 15
donorHB	2	2	1	1	1	0	1	2	0.0 to 6.0
accptHB	4	4.75	4.75	4	3.25	4	4	4.75	2.0 to 20.0
volume	929.3	1004.1	1055.4	981.8	906.5	1032.8	982.3	991.4	500.0 to 2000.0

*^a^* Flavonoids exhibiting IC_50_ values lower than 11.0 μM. *^b^* Recommended values, #star: number of property values that fall outside the 95% range of similar values for known drugs, QPlogBB: predicted brain/blood partition coefficient, QPPCaco: predicted human epithelial colorectal adenocarcinoma cell line permeability in nm/s, QPPMDCK: predicted Madin-Darby canine kidney permeability in nm/s, QPlogKhsa: predicted binding to human serum albumin, QPlogPo/w: predicted octanol/water partition coefficient, QPlogKp: skin permeability, QPlogS: predicted aqueous solubility, #metab: number of likely metabolic reactions, % HOA: predicted human oral absorption from 0 to 100%, PSA: Van der Waals surface area of polar nitrogen and oxygen atoms and carbonyl atoms, SASA: total solvent accessible surface area, MW: molecular weight, #rotor: number of non-trivial, non-hindered routable bonds.

## Data Availability

Data is contained within the article or Appendix A.

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
