# Peer review of "Flavonoids from Piper Species as Promising Antiprotozoal Agents against Giardia intestinalis: Structure-Activity Relationship and Drug-Likeness Studies"

_pharmaceuticals, 2022, doi:10.3390/ph15111386_

Round 1

Reviewer 1 Report

Article on “Flavonoids from Piper species as Promising Antiprotozoal Agents against Giardia intestinalis. Structure-Activity Relationship and Drug-Likeness Studies” may be accepted only after major revision.

Comment 1: Authors have isolated compounds that include chalcones and flavonoids. However, details of structural information for these compounds are missing except for the few chalcones and flavonoids 5 and 6

Comment 2: Exact purpose of the research was missing. Further, nowhere authors have discussed about the exact confirmation of the structures of compounds (7-29).

Comment 3: Please up load 1H, 13C nmr spectra that includes 0 delta to 14 delta value. Clarity can be increased for all the spectral images.

Comment 4:  Synthesized compounds can be further confirmed by mass spectra. Include mass spectral data in supplementary information.

Comment 5: Authors have mentioned that fifty sub-fractions, which were combined on the basis of their TLC profiles, and purified by preparative TLC to give the pure compounds (1-5 and 7-24). If possible, include Preparative TLC separation image in supporting information.

Comment 6: Is it necessary to do the biological activity of compounds (1-24) If there is no exact confirmation for compounds (7-24)

Comment 8: Abstract can be modified for better understanding.

Comment 9: Clarity of the spectral images can be increased.

Comment 10: Clarity of the figure:3 can be increased.

Comment 11: Manuscript has to be rewritten for better understanding. Refer following articles.

Ref: Barbosa E, Calzada F, Campos R. In vivo antigiardial activity of three flavonoids isolated of some medicinal plants used in Mexican traditional medicine for the treatment of diarrhea. J Ethnopharmacol. 2007 Feb 12;109(3):552-4. doi: 10.1016/j.jep.2006.09.009.

Calzada F, Cervantes-Martínez JA, Yépez-Mulia L. In vitro antiprotozoal activity from the roots of Geranium mexicanum and its constituents on Entamoeba histolytica and Giardia lamblia. J Ethnopharmacol. 2005 Apr 8;98(1-2):191-3. doi: 10.1016/j.jep.2005.01.019.

Author Response

REVIEWER 1

Comment 1: Authors have isolated compounds that include chalcones and flavonoids. However, details of structural information for these compounds are missing except for the few chalcones and flavonoids 5 and 6.

Answer. Regarding this point, I would like to highlight that we follow the international journal protocols in which the structural information of known natural products is omitted, and only it is mandatory to include the corresponding reference in which their spectrometric and spectroscopic data are reported. Thus, a paragraph regarding this point is already included in Results and Discussion section (Page 6, lines 180-191) in the current version of the manuscript. Moreover, all data (Tables 1and 2, in the Experimental section and Supporting Information, S1-S30) for the new described compounds are already included in the current version of the manuscript.

Comment 2: Exact purpose of the research was missing. Further, nowhere authors have discussed about the exact confirmation of the structures of compounds (7-29).

 Answer. Concerning this remark, the chemical structures of all the isolated flavonoids (1-29) were established by performing a comprehensive analysis of spectrometric and spectroscopic data, in particular by means of 1H and 13C NMR spectroscopic studies, including homonuclear and heteronuclear correlation experiments, and comparison of their spectroscopic data with those reported in the literature. The structural elucidation of new flavonoids (1-6) have been discussed in Results and Discussion section (Pages 2-6, lines 76-179), whereas for those compounds already reported (7-29) the corresponding reference in which their spectrometric and spectroscopic data are reported is included (Page 6, lines 180-191).

Comment 3: Please up load 1H, 13C nmr spectra that includes 0 delta to 14 delta value. Clarity can be increased for all the spectral images.

Answer. Answer. All the spectra of 1H NMR and 13C NMR included in the Supporting Information have been revised, and they are already correct.

Comment 4:  Synthesized compounds can be further confirmed by mass spectra. Include mass spectral data in supplementary information.

Answer. The mass spectra of the new described flavonoids have been included in the revised Supporting Information (Figures S5, S10, S15, S20, S24 and S30).

Comment 5: Authors have mentioned that fifty sub-fractions, which were combined on the basis of their TLC profiles, and purified by preparative TLC to give the pure compounds (1-5 and 7-24). If possible, include Preparative TLC separation image in supporting information.

Answer. Regarding this point, TLC and PTLC are It has been impossible for us since we do not have images, they are destructive techniques, due to the need to sprayed with H2O-H2SO4-AcOH (1:4:20) or to scrape to recover the compounds.

Comment 6: Is it necessary to do the biological activity of compounds (1-24) If there is no exact confirmation for compounds (7-24).

Answer. Regarding this remark, we have followed a bioassay-guided fractionation protocol to identify the bioactive constituents from Piper species, thus the biological assay of the isolated flavonoids (1-29) was carried out. Furthermore, the chemical structures of the all flavonoids (1-29) were established by performing a comprehensive analysis of spectrometric and spectroscopic data, specially by means of 1H and 13C NMR spectroscopic studies, and comparison of their spectroscopic data with those reported in the literature.

Comment 8: Abstract can be modified for better understanding.

Answer. The abstract has been corrected in the revised manuscript (Lines 29-31).

Comment 9: Clarity of the spectral images can be increased.

Answer. The spectral images have been corrected in the revised Supporting Information.

Comment 10: Clarity of the figure:3 can be increased.

Answer. This item has been corrected in the revised manuscript (Page 9, lines 319-327)

Comment 11: Manuscript has to be rewritten for better understanding. Refer following articles.

Ref: Barbosa E, Calzada F, Campos R. In vivo antigiardial activity of three flavonoids isolated of some medicinal plants used in Mexican traditional medicine for the treatment of diarrhea. J Ethnopharmacol. 2007 Feb 12;109(3):552-4. doi: 10.1016/j.jep.2006.09.009.

Calzada F, Cervantes-Martínez JA, Yépez-Mulia L. In vitro antiprotozoal activity from the roots of Geranium mexicanum and its constituents on Entamoeba histolytica and Giardia lamblia. J Ethnopharmacol. 2005 Apr 8;98(1-2):191-3. doi: 10.1016/j.jep.2005.01.019.

Answer. Regarding this comment and following the reviewer’ suggestion, We have rewritten and included some references in the revised the manuscript (Page 9, lines 316-318).

Reviewer 2 Report

The manuscript under review is highly attractive, comprehensive and well-written.

I suggest a few changes to be made, please.

- Highlight SAR of the potent hit in the abstract section showing novelty.

- Add a brief but comprehensive about the general pharmacokinetics of flavonoids with the latest citations. 

- Please change the references 18-20 with more recent ones. 

- Few of the references could be replaced with recent references 

Author Response

REVIEWER 2

Comment 1. Highlight SAR of the potent hit in the abstract section showing novelty.

Answer. The reviewer's suggestion is very interesting. This item has been corrected in the revised manuscript (Page 1, lines 29-31)

Comment 2. Add a brief but comprehensive about the general pharmacokinetics of flavonoids with the latest citations.

Answer. We thank to the reviewer for this comment, and this item has been corrected in the revised manuscript (Page 11, lines 383-388).

Comment 3. Please change the references 18-20 with more recent ones.

Answer. This item has been corrected in the revised manuscript, and references 18-20 have been replaced by more recent ones.

Comment 4. Few of the references could be replaced with recent references.

Answer. Regarding this point, references 3 and 21 have been replaced in the revised manuscript by more recent ones.

Round 2

Reviewer 1 Report

This manuscript is now recommended for acceptance.